# Encoded Library Technologies as Integrated Lead Finding Platforms for Drug Discovery

**DOI:** 10.3390/molecules24081629

**Published:** 2019-04-25

**Authors:** Johannes Ottl, Lukas Leder, Jonas V. Schaefer, Christoph E. Dumelin

**Affiliations:** Novartis Institutes for Biomedical Research, 4056 Basel, Switzerland; johannes.ottl@novartis.com (J.O.); lukas.leder@novartis.com (L.L.); jonas.schaefer1@novartis.com (J.V.S.)

**Keywords:** Encoded Library Technologies, DNA-encoded libraries, mRNA display, drug discovery, lead generation, integrated lead finding, high-throughput screening

## Abstract

The scope of targets investigated in pharmaceutical research is continuously moving into uncharted territory. Consequently, finding suitable chemical matter with current compound collections is proving increasingly difficult. Encoded library technologies enable the rapid exploration of large chemical space for the identification of ligands for such targets. These binders facilitate drug discovery projects both as tools for target validation, structural elucidation and assay development as well as starting points for medicinal chemistry. Novartis internalized two complementing encoded library platforms to accelerate the initiation of its drug discovery programs. For the identification of low-molecular weight ligands, we apply DNA-encoded libraries. In addition, encoded peptide libraries are employed to identify cyclic peptides. This review discusses how we apply these two platforms in our research and why we consider it beneficial to run both pipelines in-house.

## 1. Introduction

Advances in disease characterization and target identification [1,2] constantly move the drug discovery portfolio into unchartered territory. More and more emerging targets are attractive from a pathological point, but are at the same time challenging to modulate with established low molecular weight (LMW) compounds or biologic agents. One way to cope with that is to considerably enlarge the chemical space available to current hit finding methods (Figure 1). This is underlined by the fact that the size of compound libraries is normally limited to a range of about 10^6^ mostly due to archive and screening logistics. This lowers the chances to identify hits with high affinity and selectivity or for demanding targets. LMW agents originating from traditional compound libraries may in principle be bioavailable, allow for cell penetration and are suited to bind in defined pockets of proteins such as enzymes. Yet they frequently perform worse against protein-protein interactions (PPIs). PPIs tend to have rather large and extended interaction surfaces. On the other side of the spectrum, biologics are better suited for such applications. They are able to address extended PPIs on large surfaces and are usually both very selective and potent. However, their large molecular weight, amongst others, largely excludes addressing intracellular targets in a therapeutic manner as well as oral administration. 

Encoded libraries can help to overcome these shortcomings. Compounds originating from DNA-encoded libraries (DEL) resemble LMW molecules, but offer the opportunity to screen up to 10^9^ and more possible chemical entities. This size thus is 1’000-fold larger than a usual compound collection of pharmaceutical companies and significantly increases the chances of finding new chemical starting points. As a second pillar, encoded peptide libraries as utilized in the Novartis-internal Peptide Discovery Platform (PDP) can fill the gap between LMW and biologics. Peptides with a typical length of 5–20 amino acids have an intermediate size and are able to interact with target proteins in many ways, like covering large surfaces or small binding pockets. Depending on the set-up, peptidic libraries can encode up to 10^12^ different sequences, allowing for identification of high affinity binders against almost any target [3].

## 2. DNA-Encoded Libraries

First theoretically proposed by Brenner and Lerner in 1992 [4], the selection of LMW compounds from DNA-encoded libraries by now is an established technology employed by numerous academic and industrial laboratories [5,6,7]. Being a scalable benchtop technology for performing affinity-based hit finding, DEL screens facilitate the deep sampling of chemical space by being able to handle very large LMW libraries in a one-pot reaction. Testing billions of compounds for binding at once obviously offers benefits regarding the efficiencies and costs associated over classical high-throughput screening (HTS) formats. As the name implies, each member of a DEL is covalently connected to a unique DNA barcode which enables the unambiguous identification of retained compounds at the end of the selection process [8]. Similar to other selection systems like phage or ribosome display [9,10], this tag comprises the common “genotype-to-phenotype linkage”. For DEL it does not encode the biosynthesis of the affinity reagent itself, but rather contains instructions for their chemical synthesis. At the end of the selection process, these oligonucleotides can be PCR-amplified and the enriched compounds unambiguously identified by Next Generation Sequencing (NGS). This grants a detailed picture of the binding chemical matter, both in respect to the relative abundance of identified hits as well as to their structural compositions.

There are hundreds of different proprietary DELs used by the individual labs and institutions. Despite their distinct designs, most of them employ the same common “split-and-pool” approach for the generation of chemical diversity on a massive scale [11,12]. This technique employs a combinatorial assembly of two to four building blocks [13,14] rather than the most straightforward way of synthesizing DELs by simply tagging individual pre-made compounds. After each chemical synthesis cycle (during which also the respective part of the DNA barcode is ligated to the already existing stretch), all products are pooled together and subsequently again split in the number of new reaction containers necessary to host all building blocks of the next synthesis cycle. By employing hundreds to thousands of individual building blocks in each reaction cycle, this approach results in very large LMW compound libraries ranging from tens of thousands to hundreds of million compounds (see Figure 2A).

By combining innovative library design based on the constant development of new chemistries [15,16,17,18] with carefully crafted building block collections [19], DNA-encoded libraries are largely comprised of members with favorable drug-like properties, both regarding their molecular and physical chemistry features. In addition, opposed to some hits selected from classical HTS formats, which partially resemble natural product derivatives, DEL-derived leads all have the benefit of being mostly accessible due to the well validated synthesis procedures. These features ensure a generally high developability of molecules identified in DEL screens. 

The standard DEL selection procedure has been reviewed numerous times in the past [7,20,21], thus only the essential steps are mentioned in the following (see Figure 2B). Upon incubation of the DEL with the desired target protein, these complexes are immobilized on solid support. Alternatively, targets can be immobilized on the matrix prior to this incubation. After the non-binding compounds have been washed away, the remaining barcoded molecules are eluted by e.g., incubations at higher temperatures. After a recapture step to deplete matrix binders and any detached protein, the supernatant containing the target-interacting compounds can either be used as input material for the next round of target panning or finally be PCR-amplified for NGS analyses. It is important to keep in mind that—opposed to many other selection systems, including the one employed by PDP—for most DEL technologies library regeneration based on the enriched DNA-sequences is not possible. Thus, it is essential to have many thousands of copies of each member present in the initial starting library—otherwise, no descriptive enrichment analysis is possible after the typically performed two to three rounds of DEL selection [22]. 

Over the past decade, various developments in the DEL field have caused this technology to become a well-established tool that requires little project-specific assay development. Therefore, DEL screens provide a time- and cost-effective entry point into new projects, supporting aspects from target validation to hypothesis testing and hit discovery. Once nanomolar quantities of the purified protein of interest (POI) are available, it is straightforward to assess the general ligandability of the target. As the DEL assay format allows simultaneous screening through parallel automation with multi-channel pipettes or alternative liquid handlers [23,24], various different constructs can be tested for their suitability for drug discovery (e.g., different POI fusions, orientation of required tags, full-length protein vs. isolated domains etc.). The gained results often enable the prioritization and de-risking of projects, as well as optimization of the invested resources. In addition, through comparison with ligand-blocked selections, the parallelization permits identifying compounds that bind to specific epitopes or which comprise dedicated mode of actions (MoAs) through different set-ups.

## 3. Peptide Discovery Platform

Peptidic ligands offer an attractive alternative to LMW compounds. A comprehensive overview of the various technologies to identify peptides can be found in recent reviews [25,26]. All these methodologies share a common set-up: generation of peptide libraries, selection of peptides with desired feature(s) and finally determination of the amino acid sequences of the enriched ligands. In the early days of peptide selections, the employed libraries contained mixtures of chemically synthesized peptides (which were pooled and split) and hits binding to targets were identified by mass spectroscopy and/or Edman sequencing [27]. One key feature of such libraries is their enormous theoretical chemical diversity by including various non-natural amino acids in the synthesis. However, due to synthetic and analytical constraints, the size of such libraries is limited to about 10^7^ [28]. 

To increase the variety of peptides, molecular biology based selection approaches were introduced, linking the phenotypic (peptide sequence) to the genetic (coding) information provided by the nucleic acid sequence. The scope of phage display techniques [9], best known for antibody discovery, was expanded to select for smaller peptides and resulted in a diversity of up to 10^9^ sequences [29,30]. Phage display based on *Escherichia coli* is generally limited to the incorporation of the 20 natural amino acids and cyclization is mainly possible through disulfide bridges only. However, stably cyclized peptides are often favorable to e.g., prevent proteolysis. In addition, ring closure reduces conformational freedom and thus results in peptides with higher affinity and selectivity compared to their linear counterparts [31]. These limitations markedly decrease the accessible chemical space and a method combining the advantage of the diversity of LMW libraries and sequence multitude of genetically encoded approaches is desirable. Approaches to enlarge the chemical variety by supporting screening of stable macrocycles were for instance introduced with the “Split-Intein Circular Ligation Of Peptides and Proteins” (SICLOPPS) technology [32,33,34], employing intein fusion constructs adjacent to the peptide to enable a covalent and stable head-to-tail cyclization. Other methods use chemical or post-translational modification of peptides for stable cyclization [35]. 

A further improvement was achieved by cell-free display techniques, which dramatically increased the library size from 10^9^ to 10^13^ possible sequences, as libraries are translated *in vitro*, making the diversity limiting step of cell transformation unnecessary. Since the encompassment by phage particles is lacking, the linkage between expressed peptide and nucleic acid must happen during the translation process in order to ensure an unambiguous linkage. One of the most widespread approaches to fulfill this requirement is mRNA display, in which the antibiotic puromycin is attached to the mRNA strand and leads to a covalent link between the mRNA (Figure 3C) [36,37]. Alternatively, a short DNA oligonucleotide hybridizing to the encoding mRNA can carry the puromycin entity [38]. During the *in vitro* translation process, the freshly synthesized peptide is transferred onto the puromycin while the ribosome stalls at the vacant stop codon. Similar to other display technologies, affinity panning with an immobilized target is the preferred option to find binders. For consecutive selection cycles, the mRNA is usually converted into DNA by reverse transcription, which is amplified by PCR before initiating the next selection cycle by *in vitro* transcription and translation. Once increased recovery of the library is observed, DNA sequencing directly reveals the sequences of the binding peptides. 

To cover a maximal chemical space of displayed peptides, the so-called “reprogramming” of the genetic code, i.e., enabling the incorporation of non-natural amino acids, was the last key advancement towards the versatile Peptide Discovery Platform as it is used now at Novartis [39,40]. Traditionally, suppression of stop codons was utilized to introduce non-natural amino acids, allowing only quite limited chemical matter to be added [41,42]. A more sophisticated approach for reprogramming was enabled by two major technological advancements: Firstly, a fully reconstituted *in vitro* translation (IVT) system such as the “Protein synthesis Using Recombinant Elements” (PURE) was introduced by the Ueda lab from the University of Tokyo [43]. This complex mixture contains all necessary LMW ingredients (i.e., tri-nucleotides, salts, buffers etc.) as well as macromolecular components (like ribosomes, soluble translation factors, amino acyl tRNA synthetases (ARS), nucleic acids, etc.), which permits tailoring the composition of all required elements. For example, certain ARSs can be omitted in the mixture, thus allowing the introduction of tRNAs which have been pre-charged by acylation with unnatural amino acids. Secondly, the group of Hiroaki Suga engineered the so-called flexizymes, which conjugate any tRNA to different amino acids [44]. These specifically designed ribozymes are able to recognize a large variety of amino acids with an activated carboxyl group and transfer them efficiently to the 3′ end of the tRNAs (Figure 3A). The promiscuous behavior of the flexizymes towards amino acids and tRNAs allows the extensive reprogramming of the genetic code (Figure 3B) [45,46]. By joining the flexible IVT system (such as PURE) with both tRNAs carrying a wide diversity of amino acids charged by flexizymes and the mRNA display technology, the so-called “Random Peptide Integrated Discovery” (RaPID) system was created [3,47]. This set-up combined the main advantages of chemical diversity beyond the naturally occurring amino acids and the high number of possible sequences given by the genetically encoded selection procedure.

As described above for DEL screens, consecutive selection rounds based on the affinity of the library members to the POI are also applied with the peptide discovery technology (Figure 3C). In contrast to DEL, more iterative selection rounds are typically performed with PDP to obtain converging and enriched peptide sequences [3]. The reasons for this are discussed in more detail in the following section.

## 4. Advantages and Challenges of Encoded Library Technologies

Both DEL and PDP screens offer various benefits over conventional approaches like HTS for LMW compounds or phage display for peptide discovery, thereby enabling rapid access to both tool compounds as well as suitable starting points for drug discovery projects. First of all, the sizes of the respective input libraries are larger, allowing the chemical space to be more deeply sampled in just one screen. Additionally, in comparison to other display systems (like phage or yeast), less library focusing is inherent to these cell-free techniques, i.e., less reduction of the starting pool’s diversity due to different growth rates of the employed cellular systems. 

Due to the activity-agnostic characteristic of the ELT technology, chemical matter for diverse MoAs can be identified with only limited project-specific assay development. As the screens are neither focusing on the target’s known biochemical activity nor its interaction partners, their output can also include “silent binders”. Such binders might be of great interest as tool compounds or be employed in projects focusing on applications other than classical drug discovery, like protein degradation or peptide-drug conjugates [48,49].

Because of the combinatorial nature of the employed libraries, DEL hits usually can be identified on different “levels”: Often, not only individual ligands are identified, but rather larger families can be detected, in which related building blocks or combinations thereof are enriched [6]. This feature offers valuable insights into different structure-activity relationships (SARs), unveiling the underlying chemistry in charge of the specific binding events and thereby offering promising starting points for subsequent hit optimizations by medicinal chemistry. The same holds true for PDP where the appearance of different peptide sequence motifs in NGS analyses allows the identification of the key residues interacting with the respective target and greatly facilitates altering the properties of the identified peptidic hits. 

The fact that only limited assay development and rather low amounts of reagents (regarding both targets and libraries) are required often makes ELT screens the methods of choice for the first round of identifying binders in the pharmaceutical industry, particularly when tackling novel targets. Another key benefit of ELT is the ability to perform several selections in parallel: by simultaneously screening against multiple targets, different protein constructs and/or various selection conditions, different MoAs can be probed in a cost- and time-effective way (and not in a linear flowchart as most often done in standard HTS projects). 

As any selection system, ELTs also hold certain challenges and limitations. Direct functional read-outs can only be implemented after laborious technical development, mainly for two reasons: On the one hand, in both selection systems the respective targets are present in great excess over the individual ligands, making it challenging for enzymatic assays to give rise to tangible results. On the other hand, both in DEL and PDP screens enrichment is driven by the binding affinities and kinetics of their library members to the respective target and not based on functional effects. Thus, identified hits generally need to be thoroughly validated in subsequent assays, which requires them to be resynthesized in an untagged format in sufficient quantities. This hit re-synthesis is also important to unambiguously identify the interacting chemical matter, as despite thorough synthesis validations, the occurrence of side reactions in the library preparation cannot be fully excluded. Several methods were developed to enable the facile validation of large numbers of hits on DNA [50,51]. Nonetheless, the binding event ultimately needs to be characterized in absence of the oligonucleotide tag present in the selections to exclude possible influences on the detected binding event e.g., through charge-charge interactions. Since off-DNA hit re-synthesis requires additional resources, complete testing of rich hit lists which can contain >10,000 individual hits is virtually impossible. 

Generally, the barcodes of the library members impose an additional multitude of limitations on ELTs. Most obvious is the scope of chemical reactions compatible with DNA: even though multiple new chemical reactions are reported regularly, some conditions remain challenging, e.g., those containing strong acids, requiring extensive UV-light usage or demanding water-free conditions (although some work-arounds have been reported [52]). For PDP, the scope of bond forming reactions reported is largely limited to peptides. A noteworthy exception is the recent description of enzymes modifying the peptide backbone post-IVT [53,54]. Additional considerations include the tag’s impact on the compound’s solubility. While its increase due to DNA-conjugation is generally advantageous, this effect may require some extra attention when testing the compounds after re-synthesis. Further, the oligonucleotide may also directly interact with the target, making selecting binders against proteins with an intrinsic DNA-/RNA-binding functionality rather challenging. Such screens can still be performed—however they require thorough optimizations of the screening conditions e.g., through the addition of blocking oligonucleotides to the selection buffers or working with specific protein mutants. Another downside of the tag is its sheer dimension: Considering the size of the attached oligonucleotide in mind (up to ~100-fold bigger than the attached compound), it is not surprising that some target-ligand interactions are affected by steric hindrances. For some DELs, this can be partially overcome by *de facto* inverting the synthesis procedure and attaching the DNA to the other end of the tagged ligand. However, very deep or fully enclosed pockets may remain out of reach. An overview of all mentioned advantages and challenges for encoded library technologies in comparison to conventional HTS is provided in Table 1.

When comparing the two technologies with each other they both offer their own advantages. In the case of PDP, library generation by IVT can be accomplished in less than one hour once all reagents are available and hence can be performed at each round of selection. In contrast, DEL synthesis needs significant effort to generate the desired size of 10^9^ individual compounds. Once available, these libraries can be used for hundreds to thousands of selection campaigns. A DEL screen typically consists of two to three rounds of panning which can be completed in a few days. Together with sample preparation for sequencing, an entire DEL screen can be conducted in about one week. In contrast, a PDP screen typically takes several weeks as both an individual selection round is more labor intensive and more rounds need to be conducted (up to 10). The reason for this greater cycle number is based on a number of key differences between the two technologies. Firstly, for PDP the theoretical library size based on the degenerate code is frequently greater than the main limiting factor in the entire mix: the number of active ribosomes, which is estimated to be around 10^12^–10^13^. Consequently, each peptide sequence is statistically present as a single instance in the input library. Thus, binders can only be identified after several rounds of selection cycles. Secondly, the DNA sequences are amplified and the peptide library is regenerated at every selection cycle. This allows positive binding sequences to be enriched over consecutive cycles while keeping the total amount of library constant. This is a clear advantage over DEL where ligand enrichment takes place at the expense of total library material. As a result, PDP selections can readily be run until individual peptide sequences dominate the pool. Additionally, this enables step-wise altering the selection conditions between the rounds or revisiting earlier rounds if required. In analogy to DEL, the procedure itself can be streamlined and at least partly automated on pipetting systems which permits the screening of several targets and/or selections conditions in parallel [55]. This enables probing for specific binding epitopes or preferred MoAs during affinity panning. Another technological difference between the two platforms is that only DEL enables general pooling of libraries. For PDP, pooling is limited to libraries based on the same codon table as otherwise library resynthesis becomes impossible.

As will be discussed in detail in the following section a major difference is in the application scope of the identified ligands. DEL screening hits tend to have affinities in the low µM range. However, they readily serve as suitable starting points for further medicinal chemistry efforts. In contrast, the peptidic hits identified through PDP usually exhibit binding in the nM range. This increased affinity can be rationalized due the larger interaction surfaces of those ligands as well as the greater number of library members of about 10^12^. In spite of their potency, peptide hits are rarely applied as immediate starting points for intra-cellular targets due to their low cell-penetration although some possibilities to tweak their properties in the desired direction exist [56]. Alternatively, the identified peptides offer insights for the rational design of small moelcules through extensive medicinal chemistry efforts, converting peptide hits into LMW compounds with the desired pharmacological properties that mimic the key interacting side chains [57,58]. Only when the target is an extracellular or a membrane protein is a peptide hit more or less directly used as a chemical starting point.

## 5. Applications of Encoded Library Platforms in Drug Discovery

Major pharmaceutical companies have multiple hit finding technologies at their disposition when initiating drug discovery programs. Several factors determine which of these technologies are applied in which order to a given target. A more complete overview of all considerations will take place at a later stage of this review. When deciding between the two encoded library platforms discussed in this review, one important scientific criterion to consider is the purpose of the ligand being sought. Most DELs are designed to contain a large fraction of LMW molecules within a favorable property space for initiating medicinal chemistry campaigns towards orally bioavailable drugs. In a first pass, ligands identified from DEL screens are thus geared towards the same target application scope as hits from a similarly tailored compound collection used for HTS campaigns. Consequently, screening of DELs is the preferred option over peptide libraries if a molecule with more drug-like properties is the immediate aim of the experiment.

Two examples recently reported by GSK for receptor-interacting serine/threonine-protein kinase (RIPK1) [59] and soluble epoxide hydrolase (sEH) [60] showcase how hits from DEL screens can be progressed towards clinical candidates. In the case of RIPK1, initially conducted HTS campaigns identified potent inhibitors with different liabilities in terms of physico-chemical properties or oral exposure. A following DEL campaign identified a highly potent inhibitor, which could readily be developed into a clinical candidate for psoriasis, rheumatoid arthritis and ulcerative cancers after exchanging a central isoxazole with a triazole (Figure 4A). In the case of sEH, an initial DEL screen enriched a large number of compounds, which were triaged by reducing the amount of target immobilized. This readily gave rise to several potent inhibitors. One of these series was subsequently further optimized to a clinical candidate for chronic obstructive pulmonary disease (Figure 4B). Interestingly, even though considerable efforts were invested in the follow-up of these original DEL hits, in both examples the final candidate compounds closely resembled the initial ligands found in the DEL screen, underlining the importance of careful library design.

The scope of applications of DEL hits is not limited to serving as LMW starting points. Due to the nature of combinatorial chemistry, not every library member fully adheres to the desired property space aimed for when designing the DEL. Such compounds may still provide important information on challenging targets where other hit finding efforts do not lead to suitable ligands. In a recent example, a selective ligand termed BAY-850 for the epigenetic regulator protein ATPase family AAA-domain containing protein 2 (ATAD2) was identified in screens against its bromo-domain [61]. In spite of its molecular weight (MW) being above 650 Da, the molecule exhibited sufficient solubility and permeability to serve as a chemical probe [62] to further study the biological role of the target and the impact of its inhibition on cancer cell survival. 

As stated above, some of the key advantages of the DEL technology include the comparably rapid access to very large compound collections and the possibility to pool multiple libraries in one screen. This provides the opportunity to build DELs addressing certain areas of chemical space aiming e.g., at identifying tool compounds rather than drug-like hits. One common addition to any DEL portfolio is frequently referred to as “capped dipeptide” or “tripeptide library”. This synthesis procedure connects three building blocks by two amide bonds in a linear fashion. The advantages of such a set-up include the vast availability of building blocks as well as the efficiency of the bond forming reaction. Such DELs reliably deliver validated hits and can correspondingly serve the purpose of assessing general ligandability of a target. An example on how to progress such compounds was recently reported for the induced myeloid leukemia cell differentiation protein [63]. A DEL screen against this target yielded a tripeptidic hit with an IC_50_ of 2 µM. A co-crystal structure revealed that the ligand bound in a β-turn conformation, causing the two ends of the peptide to be in proximity of each other. Macrocyclization of this compound enabled the design of a highly selective inhibitor with an IC_50_ of <3 nM. DEL technology is also compatible with directly synthesizing macrocyclic libraries either by combinatorial assembly of the ring during its synthesis [64] or the decoration of a defined macrocyclic scaffold [65]. A noteworthy example describes the isolation of ligands from a DNA-encoded macrocyclic peptide library containing 2.4 × 10^12^ members composed of 4–20 amino acids. Such design in essence forms a bridge between typical DELs and PDP libraries [66].

Employing DEL screening to assess ligandability of a given target is further facilitated by the possibility of combining multiple libraries in one screen. Pooling DELs based on different synthetic strategies in one experiment allows efficient probing of the target for accessible binding pockets. Two recent reports on protease-activated receptor (PAR2) exemplify this well. Screening of DELs against a thermo-stabilized mutant of the receptor identified two ligand families binding to distinct sites on the target [67]. The first family, originating from a capped dipeptide library, showed considerable similarity to known agonists and indeed exhibited the same effect in cellular assays. In contrast, a library built around a central benzimidazole core identified a second family of ligands, which functioned as antagonists. These molecules bound to a novel pocket in the transmembrane region which communicates allosterically both with other antagonists as well as the predicted orthosteric binding site [68].

In contrast to DEL, the vast majority of hits identified using PDP are neither cell permeable nor orally bioavailable without substantial optimization. However, the platform very reliably delivers potent ligands against challenging targets or MoAs for which other hit finding methods are traditionally less successful, such as membrane proteins and PPIs. Thus, it is not surprising that these peptides routinely find applications as tools to enable drug discovery projects. Due to their peptidic nature, protein-protein-interactions represent an obvious MoA to be targeted by this class of ligands. A recent example in literature describes the identification of potent macrocyclic inhibitors of the interaction between the Zaire Ebola virus protein 24 and karyopherin alpha 5 [69]. Although this interaction is well known to be essential for pathogenesis of the virus, these ligands represent the first confirmed inhibitors and consequently may serve as tools for the development of anti-viral agents. 

An additional tool application of PDP-derived ligands is enabling structural determination of hard to crystallize proteins as well as giving rise to structures in different conformations. Noteworthy examples are two membrane transporter proteins: Firstly, screening of peptide libraries against *Pyrococcus furiosus* multidrug and toxic compound extrusion transporter identified four inhibitory macrocycles. These ligands facilitated structural studies of this protein by co-crystallization and were shown to bind to different regions of the target, locking it in an outward-facing conformation [70,71]. Secondly, an allosteric inhibitor identified in screens against the eukaryotic p-glycoprotein homolog from *Cyanidioschyzon merolae* co-crystallized with its target in an inward open conformation. Comparison with the apo-structure gave further insights into the mechanism of this transporter [72]. 

Due to their high affinity and good selectivity, PDP ligands can also serve as probes to study the function of their target. This was recently exemplified by a screen of the catalytic domain of Lysine-specific demethylase 4A, which identified ligands with novel binding modes [73]. These inhibitors provided insights into the target’s substrate selectivity and enabled cellular studies of global methylation pattern.

Despite the challenges with respect to oral bioavailability and cell penetration, ligands identified through PDP and related technologies [35,74] may also find application as therapeutic agents. Due to their macrocyclic, modified backbone, these peptides are very stable. A recent article by Yamagishi and co-workers suggests that backbone N-methylation alone increases the stability of peptides in general, but that macrocyclization is required in addition to result in a substantial increase in plasma stability [3]. Consequently, such ligands are well suited for addressing targets in the extracellular space. The group of Christian Heinis identified multiple binders against targets in that space, screening libraries containing >10^9^ bicyclic peptides [75,76]. In a series of articles, the group reported several inhibitors of coagulation factor XII with high potency and selectivity [77,78]. Both the potency and the blood plasma half-life of one of those ligands could further be improved by introducing two unnatural amino acids, underlining the benefit of incorporating such building blocks [79].

Another means of expanding the therapeutic scope of peptidic ligands can be achieved by the conjugation of functional payloads, generating peptide-drug conjugates (PDCs) [50]. In comparison to antibody-drug conjugates (ADCs), which attracted great attention in the past decades [80], PDCs may offer several advantages: The absence of a tertiary structure makes peptides more tolerant towards elevated temperatures and organic solvents, both of which may be required for the efficient conjugation to the desired toxic entity. Further, peptides are typically chemically synthesized which more readily allows for tuning of physico-chemical properties and for the introduction of orthogonal reactive groups—again with the aim to facilitate further modification. The small size of the targeting entity—PDP ligands typically have a MW of only 1–2 kDa—offers further benefits over biologics (which themselves have a MW of 10–150 kDa), such as short half-lives, increased extravasation and renal clearance. These properties are generally preferred when conjugating highly toxic payloads. Lastly, the likelihood of anti-drug antibodies emerging is reduced for small, backbone modified peptides over biologics. All these advantages are likely to compensate for the expected disadvantages in comparison to biologics, such as slightly lower affinity and selectivity. On the flip side, the peptides still benefit from generally higher affinities and selectivities of the targeting entity when compared to small molecule-drug conjugates with a comparably minor increase in MW [81,82]. However, small molecule drug-conjugates may exhibit better extravasation and increased tissue penetration [83]. Overall, the delivery of toxic payloads represents a unique opportunity for peptide therapeutics. In terms of entities to be conjugated, there are multiple opportunities including highly toxic drugs or radionuclides [84]. Two recent articles describe an application example for the delivery of siRNA. Firstly, fluorescently tagged versions of very potent, epithelial cell adhesion molecule (EpCAM) binding macrocycles were shown to more efficiently label cells under high-density conditions compared to conventional mAb staining methods [85]. Secondly, a liposomal siRNA delivery system was modified with an anti-EpCAM peptide lipid-derivative. The resulting liposome showed a more than 27-fold increase in cellular uptake in EpCAM-positive cell lines and inhibited gene expression in tumor tissue when systemically injected [86]. In analogy, conjugation of an antisense oligonucleotide to an engineered version of glucagon-like peptide-1 results in delivery of the cargo into pancreatic β-cells both *in vivo* and results in silencing of the targeted genes [87].

ELTs are most successful when applied to soluble, well-behaved target proteins. This is readily rationalized as in the most commonly applied screening methods ligand enrichment relies on the stable biophysical interaction with the target and is driven by the target concentration. Consequently, poorly structured proteins or targets adopting multiple conformations result in lower enrichment. This may also contribute to the observation that a larger number of DEL literature reports describe the identification of enzyme inhibitors binding to well defined pockets [8]. Nonetheless, the number of examples successfully identifying binders to challenging targets like PPIs [63,65,66,88,89] and MoAs such as agonists, positive allosteric modulators and antagonists for G-protein coupled receptors (GPCRs) [67,68,90,91,92] grows at an increasing rate. Key contributors to the recent success against GPCRs were advances in production of membrane proteins [93] and stabilization of specific conformations [94,95]. While locking the target in certain conformations increases the likelihood of identifying ligands with the desired MoAs, due to the nature of affinity panning not all identified compounds necessarily exhibit the desired functionality. The identification of agonistically acting molecules for GPCRs or other cell surface receptors can be particularly challenging. Methods allowing to introduce functionality into potent ligands would further increase the scope of addressable MoAs. Towards this end, Ito and colleagues reported the generation of peptide agonists by crosslinking two peptides binding to the Met receptor [96]. While the individual ligands themselves displayed high affinity to the receptor, only the connection of both peptides by appropriately chosen linkers resulted in molecules having an agonistic effect on cells.

To date, larger protein complexes as well as transcription factors are target classes, which are still hard to address with DEL and PDP. For the latter of these targets, the presence of DNA-binding domains and largely unstructured domains represent the major obstacles. For complexes, a multitude of challenges ranging from protein production, target complex stability and immobilized amounts as well as more opportunities for unspecific binding due to increased target size are conceivably responsible for the lack of success. Some of the abovementioned limitations may be overcome when using covalently binding ligands. In this case, enrichment is not primarily governed by reversible binding constants, making the identification of less frequently occurring binding events possible [97,98]. Both platforms require the target to be well behaved under screening conditions. As described above, PDP ligands offer larger interaction surfaces than DEL hits generally resulting in higher affinities. This is of particular advantage when addressing PPIs, a target class where PDP typically excels.

## 6. Encoded Library Technologies at Novartis

Novartis strategically conducts small molecule lead discovery projects in a holistic manner termed “integrated lead finding” [99]. This collaboration based concept merges various interdisciplinary lead finding aspects like fragment based drug discovery, HTS, biophysics, protein science, *in silico* methodologies and medicinal chemistry into one project framework. This enables multidisciplinary project teams to effectively react on findings that cross-fertilize all other involved disciplines. Often enough, the successful drug candidates are strongly influenced by the various efforts. As a consequence, the chemical matter is not a product of one certain technology, but the joint invention of the whole multidisciplinary team. With such a set-up, any potential competition between either the approaches, involved partners or sub-teams is transformed into collaboration.

As drug discovery is an innovative and evolving field, pharmaceutical companies constantly need to reflect on their lead discovery technology arsenal to remain effective and successful. Novel drug finding technologies are often invented and first applied in academic or biotech settings. This also holds true for DEL and PDP screening where multiple biotech and academic groups have specialized their efforts in. The PDP as described above was first commercialized by the Japanese biotech company “PeptiDream” and a dedicated DEL platform was developed by the Danish biotech company “Nuevolution”. To assess the value of both lead finding technologies, Novartis successfully collaborated with these biotech partners in a step-wise approach. Initial feasibility studies revealed the great potential and complementary strengths of both technologies. This led to project-based ligand-finding collaborations. Both milestone-driven partnerships delivered tangible chemical matter in a number of projects and the technologies were thus internalized to facilitate a broader and fully enabled deployment within Novartis’ project portfolio.

Integrated lead finding requires effective communication, hand-in-hand cooperation as well as full and timely data sharing between all involved sub-teams. This can be more challenging in external collaborations, particularly when the complexity of the projects requires iterative and multidisciplinary lead finding efforts. Therefore, our approach was to internalize both technologies in-house to effectively enable and support the Novartis project portfolio. Besides that, internalization allowed Novartis to deploy the technologies to any internal project without potential limitations of the technology partner company due to other competing agreements. Another important reason for the chosen business model was the freedom to internally build up an individualized and proprietary library portfolio that best suits Novartis’ needs. Lastly, the technologies required specific adaptations and innovations to fully fit to Novartis’ scientific requirements.

Through the internalization, these encoded technologies are fully embedded in Novartis’ lead finding efforts and are operated by teams with multidisciplinary backgrounds. The DEL and PDP platforms have strong synergies, both benefiting from a close collaboration and therefore optimally intertwine. Many of the involved technology aspects are synergistic and require similar expertise and set-up. The two screening platforms were grown within the screening sciences discipline and possess a shared set-up between the chemical biology and discovery chemistry groups—this interdisciplinary strategy is fundamental for their impact and ensures best leverage of the available in-house drug discovery and medicinal chemistry knowledge.

As discussed earlier, both technologies, DEL and PDP, offer various opportunities to augment more classical drug discovery approaches and to deliver complementing, synergizing chemical matter. For Novartis, DELs nicely expands other available low molecular weight screening decks, like those for fragment based and medium/high-throughput screening. Besides their high value for hit finding, the classical screening decks are also a fundamental source of proprietary building blocks to assemble new and innovative DELs with unique chemical matter. On the other side, resynthesized DEL compounds are continuously merged into the classical screening decks and thus made available for any future conventional hit finding effort.

In many of our encoded library projects, the screening results enable the design of novel or additional lead finding efforts. PDP has a strong track record in efficiently identifying ligands with high affinity and target selectivity. These represent valuable starting points for extracellular targets or can serve as high impact tools for structural efforts like the exploration of binding pockets or pharmacophore mapping. Another important application of PDP-derived ligands is their use as tools to identify small molecule chemical matter in competition screens. This was a significant bottleneck before encoded library technologies became available. Designing and developing assays to identify chemical matter for early projects was a sometimes both time-consuming and resource-intense effort, particularly in absence of known target ligands. Identifying low molecular target binders at Novartis was restricted to relatively resource intense biophysical methodologies like NMR [100] or MS-based affinity screening such as “Speedscreen” [101]. Key limitations were a quite high target protein consumption and a limited throughput. As a consequence, such screenings were reduced to a relatively small compound deck or compromised by screening large numbers of compounds pooled at a high nominal concentration, which can lead to artifacts due to solubility issues. In many cases, the identified hits have relative low target affinities. While this may not be an issue in fragment based drug discovery projects, such ligands are not suitable as probes in competition HTS assays. In order to efficiently use these molecules as tools in such set-ups, their binding mode and proper chemical modifications have to be deciphered. Furthermore, affinity maturation may often be required before they can be appropriately labeled and used as assay tools.

Today, we apply encoded library screening to novel targets in a way that is agnostic to already available tool compounds. Particularly PDP rapidly identifies suitable high affinity binders, possibly to multiple pockets on the individual targets. In few iterations, we can investigate key motifs responsible for affinity, e.g., by amino acid scanning. In most cases, it is straightforward to attach a fluorophore on the ligand at positions not involved in target binding—often the linkage vector to the encoding tags is readily available and suitable for tethering. Using such ligands as tools for competition assays is helpful to decipher the binding mode for chemical matter. In principle, DEL can also identify such tools, however as previously stated PDP has a higher likelihood to identify binders with the required affinity.

Particularly for early stage drug discovery projects both technologies help addressing key scientific questions associated with the biological context, the disease relevance or which sites on the target are associated with the desired functional effect. As an example, a recent DEL campaign against a target hypothesized to be involved in cardiomyopathy revealed the role of individual domains in contraction and relaxation of diseased heart muscle. Initial parallel DEL screening of multiple protein domains and their combinations indicated the relevance of specific combinations. This information was vital to confirm the target hypothesis and to design a suitable lead finding strategy and project flow chart. It also permitted the identification of ligands that showed functional activity and proof of concept in a bio-artificial muscle contractility assay. Similarly, several PDP campaigns strongly impact projects through the identification of unprecedented high affinity ligands for novel and often induced pockets. The compounds enable co-crystal structures and pharmacophore mapping for new MoAs or highlight the therapeutic potential of novel pockets due the observed functionality or selectivity. Acquiring these detailed insights can be vital to design the most appropriate project flowchart.

As described above, ELTs are agnostic to functional target activity and can thus be applied to a variety of possible drug discovery projects. Suitable encoded library screening projects are picked based on a science- and priority-driven assessment. Chemistry and biology ELT experts discuss and align with key project scientists how the strengths of both technologies can impact the individual project in the most efficient manner. This discussion is influenced both by other foreseen lead finding efforts and the overall goal of the project. Balancing this with resource distribution and Novartis’ priorities ensures that DEL and PDP screening is applied with highest impact on the portfolio in the framework of integrated lead finding. Still, for several projects both platforms are applied if the anticipated impact, the project-priority as well as scientific questions benefit from such an approach.

Project campaigns are tactically binned to minimize the timeframe from tool availability to the generation of ligand hit lists for the projects. This empowers the ELT team to identify possible ligands within 2–4 months from the beginning of wet-lab activities, including technology specific assay development, preparation and screening, sequencing, data parsing and eventually hit analysis. In the past years, through continuous technical and strategic optimizations, we have achieved a strong project impact track record: PDP delivered validated high affinity ligands in >85% of the performed projects while DEL screens identified validated chemical matter for >66% of the selections. These projects included many challenging targets with unprecedented chemical matter or those that were a challenge for encoded technologies (e.g., DNA-binding transcription factors or integral membrane proteins).

Other pharmaceutical companies successfully apply similar technologies with a different model compared to Novartis, e.g., by internalizing full biotech companies specialized on technologies like DEL, or by fully outsourcing certain project aspects (typically the screening and/or early ligand optimization) to biotech collaboration partners. Besides Novartis, several large pharmaceutical companies in-licensed encoded library technologies or have built-up their own capacities in the field. To our knowledge, however, Novartis is uniquely applying both screening technologies in a synergistic and combined set-up as an integrated approach while other players in the field only apply either DEL or PDP. In our eyes, fully embedding both methodologies into Novartis’ drug discovery teams with enabled expert scientists shows great impact and ensures effective and seamless collaborations as described above.

## 7. Future Directions

Over the last decade, ELTs emerged as powerful method to rapidly access large chemical space using comparably simple affinity screening. As stated earlier, one of the major limitations of ELTs is that the most frequently applied selection methodologies primarily enrich ligands based on binding to the target protein. While variations of the conditions allow the detection of differences in affinity and selectivity (e.g., by varying targets or their concentration) or may hint at possible binding sites (e.g., by using blocking agents), a direct functional read-out remains challenging. Still, a few examples on how to tackle this limitation were described recently: For peptide libraries, SICLOPPS enables functional in-cell selections of peptides by coupling the desired activity to the survival of the cell [102]. For small molecules, the Paegel group reported the combination of DNA-encoded small molecule on bead libraries with microdroplet-based sorting to enrich for ligands with the desired biochemical activity. After an initial model experiment [103] the group recently reported the identification of novel inhibitors of autotaxin in an activity based DEL screen [104]. Since microfluidic devices are also compatible with cellular systems [105,106], such functional selections can be expanded to the identification of cellular active compounds from both encoded technologies. This is particularly attractive for the screening of cyclic peptides for which cell penetration still represents one of the major hurdles. Correspondingly, in the upcoming years there should be more reports on functional read-outs, combining encoded libraries with miniaturized compartmentalization strategies.

Another technological aspect of these platforms to likely see major advances in the next years is data analysis. The read-outs of DEL and PDP screening—typically performed by NGS—generate large amounts of data. The reported analysis methods mainly focus on the similarity of enriched building blocks or their combinations as well as the performance of the enriched ligands under the different screening conditions [6]. However, it is conceivable that application of machine learning algorithms can both facilitate data interpretation as well as enable deeper data mining. Firstly, hit calling mostly is a rather tedious, complex process—particularly when starting with a rich hit list of multiple related selections. Secondly, the sequencing results are currently primarily used to decode the structure of the enriched binders. However, it is conceivable that the aggregated data of the occurrence of building blocks and ligands can be interpreted more deeply to inform hit resynthesis. For instance, this may allow to speed up the lead generation process by immediately designing “second-generation compounds” to jump-start medicinal chemistry based on the available structural information of the enriched features.

The application scope of ligands originating from ELT will most likely expand in the near future as well. In the recent years, a number of modalities gained attention in which molecules exhibiting different functions are chemically connected. This includes the abovementioned PDCs, but also bifunctional degraders [48]. Ligands identified through DEL and PDP are uniquely positioned to be applied to that modality space. The former attachment site of attachment to DNA can easily be chemically modified and obviously should tolerate even larger and bulkier groups without impacting ligand binding. This can be leveraged as exit vectors suitable to decorate the identified molecules with further functional entities.

These and other future optimizations will further increase the impact of encoded library technologies on drug discovery. The complementing in-house availability of the two platforms discussed in this review allows Novartis to apply them synergistically to our projects and to identify ligands against almost any target in a way complementary to more conventional HTS or fragment based methods. Due to the large chemical space covered, these platforms enable us to rapidly respond to both the evolving targets space and novel modalities in an efficient and thorough way. Thus, the full embedding of both platforms into our integrated lead finding concept allows us to broadly deploy them to the Novartis project portfolio and to maximize their impact on future drug discovery.

## Figures and Tables

**Figure 1 molecules-24-01629-f001:**
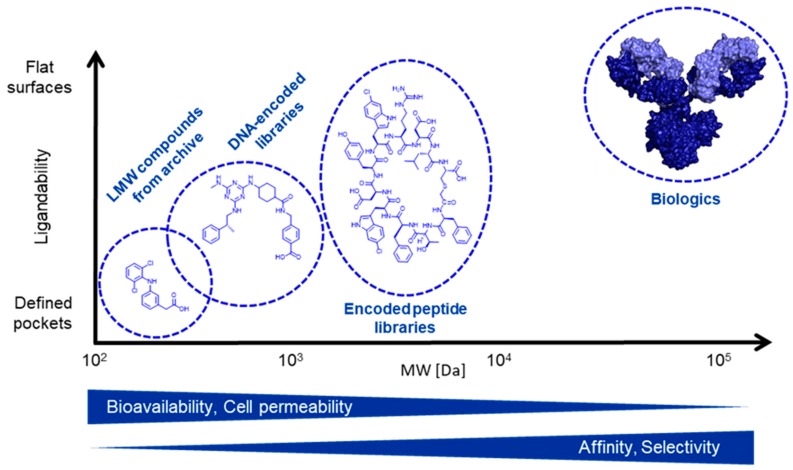
Model of chemical space available for hit finding and positioning of the encoded library technologies DEL and PDP. While those ligands on the left typically have a higher bioavailability and cell permeability due to their smaller size, this gets less and less pronounced towards larger molecular weights. In contrast, bigger molecules like peptides and biologics most often possess higher affinities and selectivities towards their targets compared to LMW compounds. The typical numbers of library sizes are about 10^6^ for archive-based compound collections, about 10^9^ for DELs, 10^12^ for peptide libraries and about 10^10^ for biological agents.

**Figure 2 molecules-24-01629-f002:**
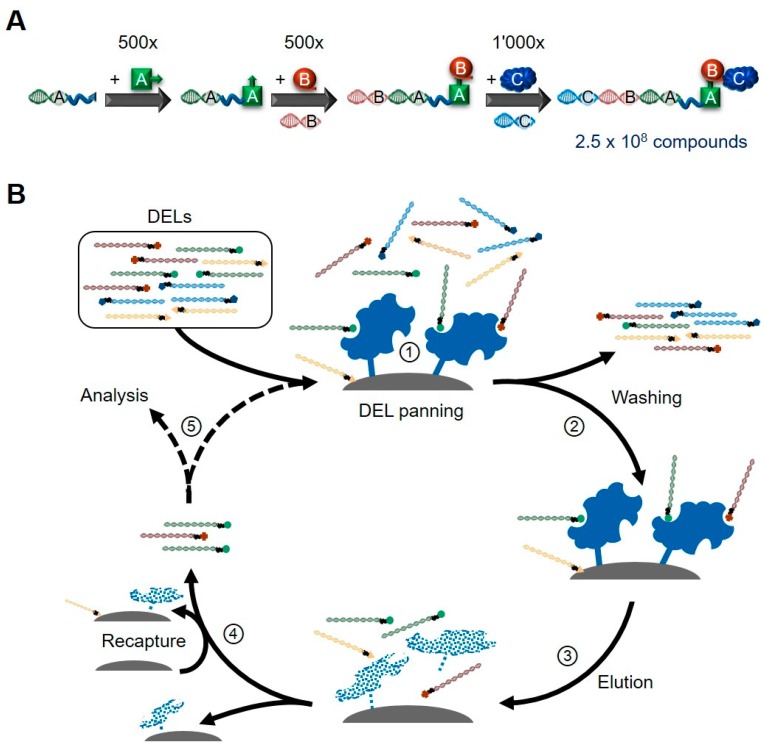
Schematic overviews of library construction and iterative DEL selection rounds (**A**) Employing the “split-and-pool” approach, very large DELs can be created. Shown is an example of combining three building blocks in parallel to the respective coding oligonucleotide fragments. (**B**) A typical DEL selection cycle consists of various steps: ① DELs are incubated with target proteins immobilized on solid support (matrix, e.g., magnetic beads). Alternatively, binding of DELs to the targets can be performed in solution prior to immobilization. ② Through various washing steps, non-binding or weakly-interacting compounds are removed and binding molecules retained. ③ Interactors are eluted from targets by specific or heat elution, breaking up the underlying interactions and partly denaturing and detaching the immobilized proteins. ④ To remove these detached proteins and compounds interacting with the matrix, a recapture step is performed with new matrix material. ⑤ The non-binding supernatant, containing the hits from the selection round, is either used for the next round of selection or as input material for PCR amplifications and analysis by NGS. Components are not drawn to scale.

**Figure 3 molecules-24-01629-f003:**
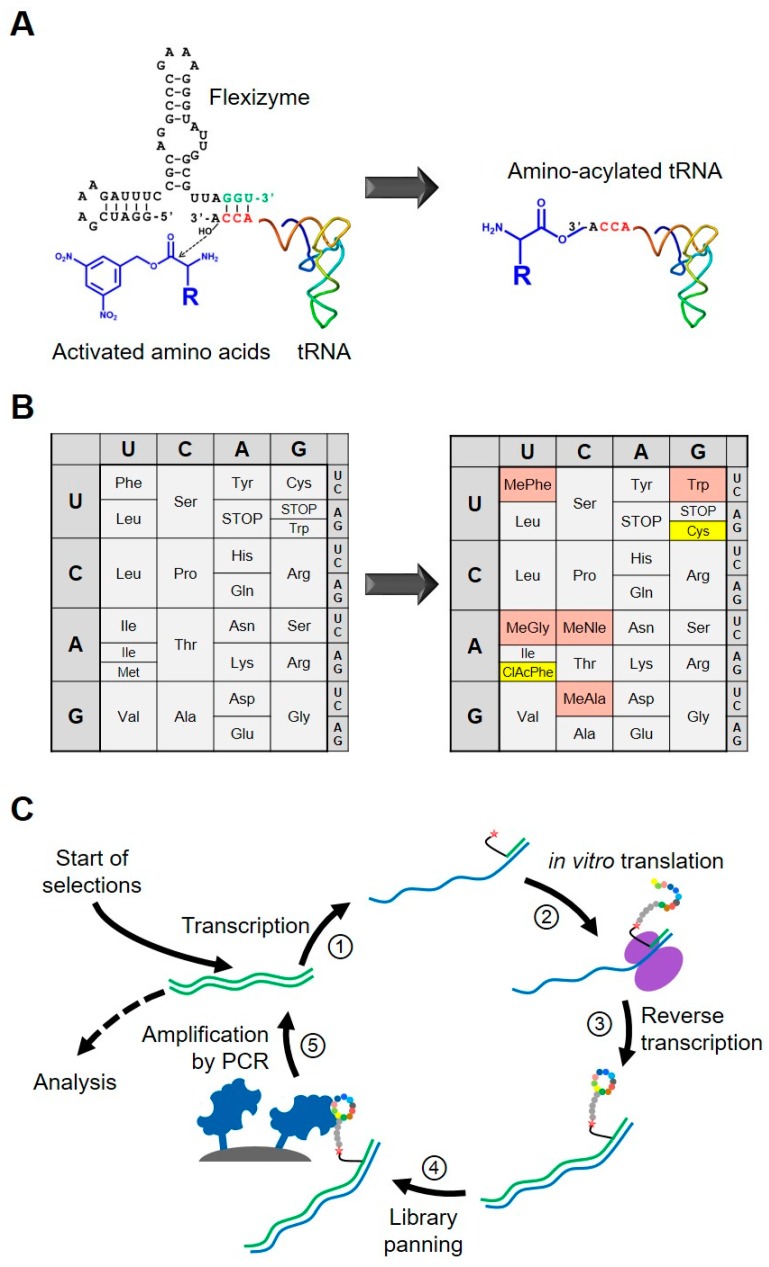
Cornerstones of the PDP technology: (**A**) Flexizyme catalyzes charging of tRNAs with a variety of natural and non-natural amino acids with an active ester at their carboxyl group. (**B**) Tailored *in vivo* translation systems (e.g., PURE) allows the reprogramming of genetic code. In the current example, the following non-natural amino acids are introduced: *N*-chloracetyl-phenylalanine (ClAcPhe) serving for the thioether based cyclization; *N*-methylalanine (MeAla), *N*-methylglycine (MeGly), *N*-methylnorleucine (MeNle) and *N*-methylphenylalanine (MePhe). (**C**) mRNA display based iterative selection process starts from the transcription of the DNA libraries into mRNA ① and its translation into the peptide and attachment to the mRNA through a puromycin based linker ②. The thioether bridge forms between the N-terminal ClAcPhe and C-terminal cysteine residue of the translated peptide. ③ Reverse transcription of the mRNA results in a more stable RNA/DNA hybrid tag and prevents RNA aptamer formation. ④ Panning of the peptide with the RNA/DNA tag on the immobilized tag enriches binding sequences. ⑤ Elution and amplification of the DNA tag yields the starting point for the next selection round. After completion of multiple selection rounds a comprehensive hit list is obtained by NGS for all rounds. Components are not drawn to scale.

**Figure 4 molecules-24-01629-f004:**
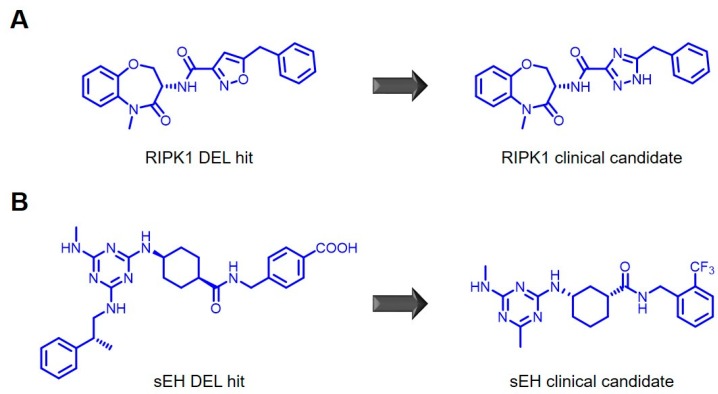
Chemical structures of initial hits derived from DEL screens and of the resulting clinical candidates after medicinal chemistry efforts for the two reported projects against RIPK1 (**A**) and sEH (**B**).

**Table 1 molecules-24-01629-t001:** Advantages and challenges/limitations of the ELT technology. Details on the key points are given in the main text.

Advantages	Challenges/Limitations
Deep sampling of chemical space by screening very large libraries Activity-agnostic screening method allowing identification of hits with diverse MoAs Informative hit enrichment, offering valuable insights into SAR Screen outputs offer rapid access to tool compounds and starting points for drug discovery projects Limited assay development required Assay format allows parallel screenings Low reagent requirement per screen Fast selection process (several selections can be run simultaneously within a few weeks)	Affinity-driven selection process; direct functional read-outs challenging Hits require resynthesis after selection for subsequent validation DNA-/RNA-binding targets challenging due to potential interactions between tag and protein Oligonucleotide tag might affect target-binder interactions and cause steric hindrance Encoding entity restricts extent of possible chemical reactions and can influence properties of ligands

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
