# Peer review of "Encoded Library Technologies as Integrated Lead Finding Platforms for Drug Discovery"

_molecules, 2019, doi:10.3390/molecules24081629_

Round 1

Reviewer 1 Report

In manuscript, authors review application of Encoded Library Technologies (ELT) in drug discovery. Authors mainly focus DNA-encoded libraries and peptide-encoded libraries. The review’s structure is clear and covers many concepts about the ELT. However, more revisions are necessary to publish this review on Molecules.

Comments

1. Authors mainly focus two techniques of DNA-encoded libraries and peptide-encoded libraries. But, what is advantage and disadvantage of them when compare each other?  Especially, in drug discovery.

2. Authors describe many details about techniques, especially about working principle.  However, as a review, recent development of the technique is very important. Authors should focus this point more.

3. Manuscript is long. Sometime, table is a good option to show different work now. For example, it is better to list a table to summary work in section 2. DNA-encoded libraries and 3. Peptide Discovery Platform. Then, some related word in manuscript can be reduced.

4. Title 7 and title 6 are repeated in manuscript.

Author Response

We highly value the constructive input by the reviewer and have adjusted the manuscript according to the suggestions. A point-by-point reply is given in the attached document.

Reviewer 2 Report

In their Review Article entitled "Encoded Library Technologies as Integrated Lead Finding Platforms for Drug Discovery" Ottl et al. thoroughly analyze and compare the impact and challenges of the two encoded lead finding platforms DNA-encoded chemical libraries and encoded cyclic peptide libraries (in general and in particularly within Novartis).

The Article is generally nicely written and gives readers in the field of drug discovery a very good overview of these emerging platform technologies in an integrated lead finding cascade.

Specific comments and suggestions:

- Line 9: suggest to leave out the article "the"

- General intro: too many repetitions of words such as "highly" and "straightforward" and too many long and nested sentences

- Line 240: Hit validation of large ensembles of hits possible and practiced "on-DNA" e.g., by SPR or fluorescence polarization experiments.

- Line 211/371: expand and add references regarding small-molecule drug conjugates (pros & cons: better extravasation and tissue penetration, shorter half-life, ...)

- Line 315: greek character missing

Line 388: make reference to Astra Zeneca's / Ionis' pancreatic beta-cells peptide targeted oligonucleotides

Paragraph 6. and 7. contains the same title: Encoded Library Technologies at Novartis; please change.

In addition, please elaborate and make reference to the recent advances in the field of functional and cell-based DEL microfluidic and droplet screening systems.

Author Response

(The authors gave the same response as above.)

Round 2

Reviewer 1 Report

Authors have improved manuscript much. I would like to recommend its publication on 

Molecules.